# Characteristics of Some Wild Olive Phenotypes (Oleaster) Selected from the Western Mountains of Syria

**Reem Abdul Hamid [1], Hussam Hag Husein [2],* and Rupert Bäumler [2]**

1 Administration of Horticulture Research, General Commission for Scientific Agriculture Research, Damascus P.O. Box 30621, Syria; reem_ahamid@gcsar.gov.sy

2 Institute of Geography, FAU Erlangen-Nuremberg University, Wetterkreuz15, 91058 Erlangen, Germany; rupert.baeumler@fau.de

\* Correspondence: hussam.husein@fau.de; Tel.: +49-9138523305

**Abstract:** This study presents the evaluation of some technological and production specifications of 20 selected wild olive (oleaster) phenotypes from Hama Province, western–central Syria. The analyses of oil quantity showed that the olive oil (OO) extracted ranged from 10.43 to 29.3%. The fatty acid composition determined by gas chromatography (m/m%, methyl esters), conforming to commercial standards, showed the percentages of palmitic (ranged 13.2–15.06%), stearic (2.27–4.2%), arachidic (0.42–0.7%), palmitoleic (0.73–1.25%), oleic (64.29–73.17%), linoleic (8.96–16.45%), and linolenic (0.23–1.6%). Our results suggest that, despite being in a harsh environment and lacking agricultural service, two wild olive phenotypes (WA4, WA6) are interesting since their fruits showed high-quality properties (fruit weight 2.16, 3.24 g; flesh 75.83, 86.2, respectively), high content of OO% (29.27, 29.01, respectively), and better fatty acid composition (oleic % 68.45, 66.74, respectively). This enables them to be a very promising introductory feature in olive genetic improvement processes. Thus, both phenotypes were adopted tentatively as inputs, the first for oil purposes and the second for dual purposes (oil and table olives). It will be important to further evaluate these promising phenotypes in terms of their OO minor compounds, as well as their ability to resist biotic and abiotic stresses.

**Keywords:** biodiversity; fatty acid composition; oleic acid; accession; genetic improvement; oleaster

## 1. Introduction

The origin of the olive (*Olea europaea* L.) remains unclear [1–5]. However, the archeological, geographical, and biological studies suggest that cultivated olives (*O. europaea* L. var. *sativa* Lehr) resulted from the domestication of wild olives [6,7], particularly oleaster (*O. europaea* L. subsp. *sylvestris* (Miller) Hegi) [8–10]. These wild forms have different morphological and biological features. Usually, they grow spontaneously in the form of thorny shrubs with small-size fruits, a higher stone/mesocarp ratio, spinescent juvenile shoots, relatively low oil content, and a longer juvenile stage [9,11–17]. Their early domestication has resulted in the development of a huge number of varieties [5]. Domestication has always been carried out through vegetative multiplication for empirically selected phenotypes with desirable agronomic traits, such as large fruit size and/or high oil content [9]. Sometimes, the direct planting of cuttings or the grafting onto wild forms for individuals presenting favorable traits has resulted in superior performance in fruit size and oil content [2,9,11,12].

Given that the olive is a long-living and evergreen species, it cannot be ruled out that individuals and genotypes reaching very far back to the origins of domestication may still be found [7]. However, the path from wild to domesticated status may have been very short and may still be occurring [18]. The first apparent phenotypic changes in domesticated olives include increases in fruit production and growth ring enlargement,

which involves a fleshy and oil-containing mesocarp. While wild olives are cross-fertilized and reproduced by seed, cultivated varieties are self-fertilized and maintained by vegetative propagation [19]. This confers wild olives a higher level of heterozygosis and genetic variability than cultivated varieties [20].

Olive oil is composed of triglycerides (98–99%) [21], and minor components such as sterols [22], waxes [23], tocopherols [24], carotenes and chlorophylls [25], phenolic compounds [26], and volatiles [27]. A balanced fatty acid composition contributes to the biological virtues of olive oil [28]. Therefore, as fatty acid is a genetic characteristic [29], many researchers use ratios of fatty acids to characterize olive varieties [30,31].

Syria ranks sixth place, on a global scale, in terms of olive production [32] and possesses very rich germplasms of olives. Wild olives (possible ancestors of the cultivated genotypes) still exist in undisturbed maquis and forests of Syria [8,10,33], besides more than seventy planted olive cultivars [34–36]. However, only five conventional cultivars (Zaity, Sorani, Doebli, Khodeiri, and Kaissy) represent about 89.3% of the total olive trees cultivated [35]. Thus, there is an urgent need to release new varieties that are more suitable for modern farming techniques. The process of genetic improvement requires a large amount of time and a high cost, in addition to the unpredictability of reaching the desired goal [37]; therefore, a selection process seems to be easier, cheaper, and faster [38]. Consequently, an ongoing program has been set up in Syria to select promising wild olive individuals [36].

This study is considered an intersection between the biodiversity project in Syria and the project of genetic improvement of olives in Syria, the latter aiming to inventory and collect the Syrian wild olive and assess their agronomic performance for future use in olive breeding programs [36].

In the current study, 20 wild olive varieties are evaluated to select superior individuals for use in genetic improvement processes. The evaluation was based on fruit specifications, quality and quantity of oil, and fatty acid compositions under Syrian conditions. The approach went through an initial selection in the habitat, and then morphological, technological, and productivity specification evaluation for the individuals. The superior individuals were re-evaluated according to the levels of oleic acid, which is a required feature for conferring the oxidative stability of the oil. The work resulted in the selection of two promising varieties of wild olives with a greater number of positive properties, whether as oil, table olives, or with a double purpose.

## 2. Materials and Methods

### 2.1. Plant Materials

Plant materials were obtained from the western mountains of Hama City, Syria, at heights ranging from 443 to 544 m.a.s.l and with the coordinates $35°02'$–$35°05'$ N, $36°20'$–$36°29'$ E, where wild and cultivated olives still coexist within pockets of terra rossa. Three mountainous sites (Sygata, Tel Afar, Kafer Nbbl) were surveyed. The area is subject to a Mediterranean climate with an average rainfall of up to 800 mm/year.

These sites possess variable and small populations of isolated trees of different ages, but, in general, medium-aged trees (20–30 years) prevail as perennial trees have almost disappeared due to over-cutting. During the two growing seasons, 20 phenotypes of wild olive (oleaster) were selected to study as promising accessions with apparent potential based on crop load, fruit size, and tree vigor. Figure 1 shows how to mark the selected trees; their locations were restricted by the Global Positioning System (GPS) and then by code.

Morphological readings of leaves, flowers, and fruits were taken corresponding to the seasonal phenology. Plant materials were collected from each tree, and morphologically described following the International Olive Oil Council [30].

For each location and tree, three samples of plant materials were collected. The fruits were hand-picked from the tree during two growing seasons, 2009 and 2010. The maturity was indicated by fruit color using the Jaen index (0–7) [39]; thus, the fruits were picked up from each coded plant when 70% of the peel and pulp converted to black, which corresponds to the (5) of the Jaen index. The healthy fruits without any morphological

biotic and abiotic stress were selected and shipped on the same day to the laboratory. Plants materials were performed in the laboratory of fruit physiology at the General Commission of Scientific Agricultural Research (GCSAR), Syria.

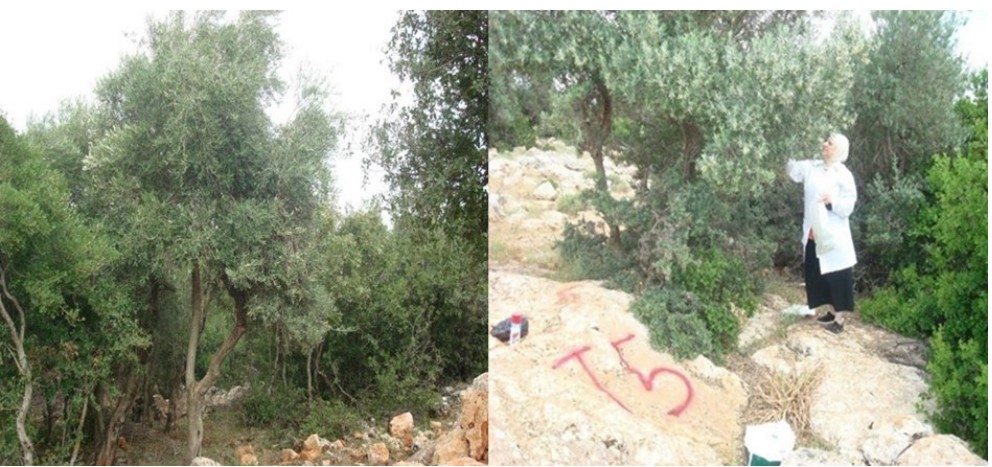

**Figure 1.** Wild olive phenotypes selection and plant material colocation.

The oil was extracted from the samples. 150 g of fruits were crushed for 30 s in a Foss-Knifetec 1095 electric mill. The resulting paste was mixed at a temperature of ca. 27 °C for 30 min, and then 8 g of the paste was placed with 100 mL of petroleum ether in the Soxhlet apparatus for 6 h (boiling point ranges from 40 to 60 °C) [40].

The extracts were centrifuged to remove insoluble substances, and they evaporated to dryness on a rotary evaporator. After evaporation of the solvent, the oils were stored in dark glass bottles. The percentage of oil was calculated as the following:

Percentage of extractable oil (*EO*):

$$EO = \frac{\text{total oil weight}}{\text{sample weight}} \times 100$$

Percentage of oil based on wet weight (*EOW*):

$$EOW = \frac{\text{weight of extracted oil after evaporation} \times \text{percent of the dry olive material}}{100}$$

The percentage of oil based on the wet material was classified into low (16–18%), medium (18–20%), and high (more than 20%) [22].

### 2.2. Determination of Fatty Acids Composition

Fatty acids were determined as fatty acid methyl esters (FAMEs) [41,42] using 0.1 g of oil samples that were vigorously mixed with 200 μL of a methanol solution of KOH (2 M) and 2 mL of n-hexane and then settled for 5 min. The top layer was injected into a gas chromatograph system coupled to a flame ionization detector (FID). A column of fused silica capillary (DB-wax, Agilent Technologies, Wilmington, DE, USA) was used. The nitrogen gas was used as carrier gas with a flow of 1.69 mL/min, the temperature was kept for the injector at 250 °C, and a split ratio of 1:50 was used. A gradient oven temperature program was adopted, with the initial temperature set at 165 °C for 15 min. Then, the temperature was raised from 165 °C to 200 °C at a rate of 5 °C/min. For 2 min, the temperature was kept at 200 °C, then increased from 200 °C to 240 °C at a rate of 5 °C/min. Finally, the temperature was kept at 240 °C for 5 min. The Authentic Commercial Standards were used to identify each FAME, and the concentration was calculated as a percentage of total peaks areas. A standard solution of FAME was prepared for fourteen fatty acids based

on standard reference solutions included in two special groups ME14-ME19 produced by (Sigma); the final concentration of fatty acids was 18 mg/L.

The iso-octane mixture was used throughout the preparation, as well as the RM6 (0-7631) standard mixture produced by (Supelco, Bellefonte, PA, USA) made of seven fatty acids (C14: 0, C18: 3, C18: 2, C18: 1, C18: 0, C16: 1, C16: 0) to help distinguish peaks of oil samples.

Separation of FAME was performed using a fused silica capillary column (60 m length, 0.25 mm internal dimension, 0.25 μm film thickness), a split injection system was used to split the injected sample, and the splitting ratio was 1:10. The nitrogen gas was used as carrier gas with a flow of 1.69 mL/min. In the injector, the temperature of the furnace was kept at 147 °C for 30 min. Authentic commercial standards were used to identify each FAME, and the concentration was calculated as a percentage of total peaks areas.

### 2.3. Statistical Analyses

The statistical program Genstat 7 was used to calculate the coefficient of variation (CV) and to evaluate the significant differences at the 0.05 confidence level. Pearson's test was used for correlation analysis. A cluster analysis was conducted on the Euclidean distance matrix based on the normalized data processed by hierarchical cluster analysis (HCA) using the XLSTAT 2012 version (Addinsoft, New York, NY, USA) [43].

## 3. Results and Discussion

Biometrics are widely considered in table olive and olive oil production [44], where the oil quantity and quality, in addition to the qualitative characteristics of the fruits (fruit weight, seed weight, flesh), are among the most important characteristics that are used to distinguish and rank cultivated or wild olive varieties. These are genetic traits affected by environmental conditions, soil, and agricultural management [45]. The quality and quantity of olive are not only related to pre-harvest factors but also to post-harvest conditions [46]. As well, the average fruit fresh weight is a crucial agronomic parameter for a preliminary selection of varieties for table olives, oil destination, or even both uses.

### 3.1. Morphological Characteristics

Table 1 shows the averages of the morphological readings of the stone, leaf, and fruit during two seasons of the selected wild olives' phenotypes. The results suggest a significant difference between the following: stone lengths, widths, and length–width ratios. The stones have different shapes with length ranges from 0.86 to 1.63 cm, and widths from 0.43 to 0.90 cm. The stone's weight ranged from (0.32 to 0.66 g). The morphological characteristics of the leaf suggest only a significant difference for the length and width.

Table 1. Morphological characteristics of stone, leaf, and fruit of the studied wild olive phenotypes (*n* = 3).

| Phenotype Code * | Stone | | | | Leaf | | | | Average Weight from 40 Fruits (g) * | Average Weight from 40 Stone (g) * | Flesh Weigt % |
|---|---|---|---|---|---|---|---|---|---|---|---|
| | Length (cm) | Width (cm) | Ratio (Length/Width) ** | Shape | Length (cm) | Width (cm) | Ratio (Length/Width) | Shape | | | |
| WS1 | 1.30 | 0.53 | 2.45 [a,b] | elongated | 3.80 | 1.10 | 3.40 | elliptic | 1.69 [d] ± 050 | 0.50 [b] ± 0.04 | 70.10 |
| WS2 | 1.45 | 0.63 | 2.30 [a,b] | elongated | 3.50 | 0.80 | 4.37 | elliptic-lanceolate | 3.13 [a] ± 0.08 | 0.58 [b] ± 0.05 | 81.49 |
| WS3 | 1.15 | 0.50 | 2.30 [a,b] | elongated | 4.20 | 1.10 | 3.81 | elliptic | 1.85 [d] ± 0.057 | 0.40 [c] ± 0.05 | 78.45 |
| WS4 | 1.33 | 0.63 | 2.11 [a,b] | elongated | 4.80 | 1.10 | 4.36 | elliptic-lanceolate | 2.69 [b] ± 0.057 | 0.47 [c] ± 0.00 | 82.23 |
| WS5 | 1.40 | 0.63 | 2.22 [a,b] | elongated | 3.60 | 0.90 | 4.00 | elliptic-lanceolate | 2.49 [b] ± 0.105 | 0.47 [c] ± 0.05 | 81.00 |
| WA1 | 1.10 | 0.60 | 1.83 | ovoid | 4.60 | 1.40 | 3.20 | elliptic | 2.25 [b] ± 0.27 | 0.54 [b] ± 0.10 | 75.83 |
| WA2 | 1.40 | 0.60 | 2.31 [a,b] | elongated | 4.50 | 1.30 | 3.46 | elliptic | 2.16 [c] ± 0.175 | 0.54 [b] ± 0.10 | 74.79 |
| WA3 | 1.30 | 0.50 | 2.60 [a,b] | elongated | 3.20 | 1.40 | 3.14 | elliptic | 2.18 [c] ± 0.06 | 0.61 [a] ± 0.10 | 71.95 |
| WA4 | 1.00 | 0.60 | 1.66 [d] | ovoid | 3.80 | 1.20 | 2.80 | elliptic | 2.10 [c] ± 0.20 | 0.51 [b] ± 0.05 | 75.64 |
| WA5 | 1.50 | 0.50 | 3.00 [a] | elongated | 4.40 | 1.50 | 3.10 | elliptic | 3.06 [a] ± 0.36 | 0.56 [a] ± 0.06 | 81.50 |
| WA6 | 1.10 | 0.60 | 1.83 [d] | ovoid | 3.40 | 1.40 | 4.14 | elliptic-lanceolate | 3.24 [a] ± 0.18 | 0.44 [c] ± 0.10 | 86.20 |
| WA7 | 1.40 | 0.70 | 2.00 [c] | elliptic | 4.7 | 0.9 | 5.11 | elliptic-lanceolate | 1.59 [d] ± 0.04 | 0.41 [c] ± 0.1 | 74.27 |
| WT1 | 0.86 | 0.43 | 2 [c] | elliptic | 5.80 | 1.10 | 4.18 | elliptic-lanceolate | 1.39 [d] ± 0.33 | 0.32 [d] ± 0 | 76.64 |
| WT2 | 1.30 | 0.60 | 2.16 [c] | elliptic | 4.60 | 1.50 | 3.06 | elliptic | 2.17 [c] ± 0.07 | 0.46 [c] ± 0.05 | 78.66 |
| WT3 | 1.55 | 0.43 | 3.60 [a] | elongated | 4.60 | 1.30 | 3.53 | elliptic | 2.70 [b] ± 0.06 | 0.66 [a] ± 0.06 | 75.60 |
| WT4 | 1.20 | 0.90 | 1.33 [e] | spherical | 6.30 | 1.20 | 5.25 | elliptic-lanceolate | 1.47 [d] ± 0.07 | 0.44 [c] ± 0.01 | 69.86 |
| WT5 | 1.30 | 0.60 | 2.16 [c] | elliptic | 5.50 | 1.50 | 3.60 | elliptic | 2.21 [c] ± 0.06 | 0.50 [b] ± 0.10 | 77.23 |
| WT6 | 1.36 | 0.56 | 2.42 [a,b] | elongated | 4.30 | 1.20 | 3.58 | elliptic | 2.99 [a] ± 0.05 | 0.39 [d] ± 0.02 | 86.66 |
| WT7 | 1.10 | 0.50 | 2.20 [c] | elliptic | 6.80 | 1.50 | 4.50 | elliptic-lanceolate | 3.04 [a] ± 0.11 | 0.61 [a] ± 0.08 | 79.75 |
| WT8 | 1.63 | 0.60 | 2.71 [a,b] | elongated | 4.70 | 1.20 | 4.00 | elliptic-lanceolate | 3.00 [a] ± 0.22 | 0.63 [a] ± 0.11 | 78.82 |
| C.V | 9.00 | 7.80 | 9.10 | - | 6.80 | 9.20 | 0.00 | - | 8.60 | 16.40 | - |
| L.S.D 0.05 | | | | - | 0.50 | 0.17 | 0.00 | - | 0.33 | 0.13 | - |

(*) The first letter refers to (wild olive), and the second letter refers to the place of existence (S = Sygata; T = Tel Afar; A = Afar). (**) The means with different letters in parentheses ([a,b,c,d,e]) for the same quality parameter indicate significant differences between the means at the 0.05 confidence level.

The great variability in the means of the average fruit weight was observed among the studied phenotypes; it ranged from 1.39 to 3.24 g. The percentage of flesh weight ranged from 69.86 to 86.66%, and all the cultivars had a flesh weight lower than 2.4 g. A very high correlation was found between the fruit weight and the flesh weight (0.990, $p < 0.001$), whereas the stone weight was less correlated with the fruit weight (0.495, $p < 0.001$), meaning that the stone had a lesser influence on the total fruit weight. The weight of the fruits was higher than that of the fruits of the cultivated variety Souri [47]. However, it is a little less than the main cultivated varieties in Syria [48].

The (WS2, WA6) phenotypes showed a maximum weight of the fruit of 3.13 and 3.24 g with flesh weights of 81% and 86.2%, respectively. The maximum flesh weight that appears in (WT6) is 86.66% with a fruit weight of 2.99 g.

The analysis of variance revealed significant differences among the olive accessions for the main studied parameters, indicating a high degree of variability among oleasters found in the same environment. As expected, the values for the fruit size in wild olives were lower than those obtained in cultivated materials, with intermediate values between 1.39 and 3.24 g; this is classified as light to medium according to the standard guide for the characterization of olives [30].

The cluster analyses of the stone (A), the leaf (B), and the fruit (C) characteristics have been presented in (Figure 2). The morphological characteristics of stones and leaves are clustered into two identical groups. The first group contains two subgroups: (WT3, WT4) and (WA7, WA1, WA6, WT5, WT2, WT1, WS3, WS2, WA2). The second group also contains two subgroups: (WT8, WT6, WA3, WS1) and (WS3, WT2, WT7, WS5, WA5, WS4). The fruits are clustered according to their characteristics in three groups. The first group (average fruit weight 2.18–1.47 g, flesh weight 71.95–69.86%), the second group contains two subgroups (average fruit weight 2.706–1.59 g, flesh weight 74.27–77.23%), (average fruit weight 3.043–1.8 g, flesh weight 82.23–78.78%, and the third group (average fruit weight 3.246–3.0 g, flesh weight 86%). The last subgroup seems to be very promising as table olive varieties.

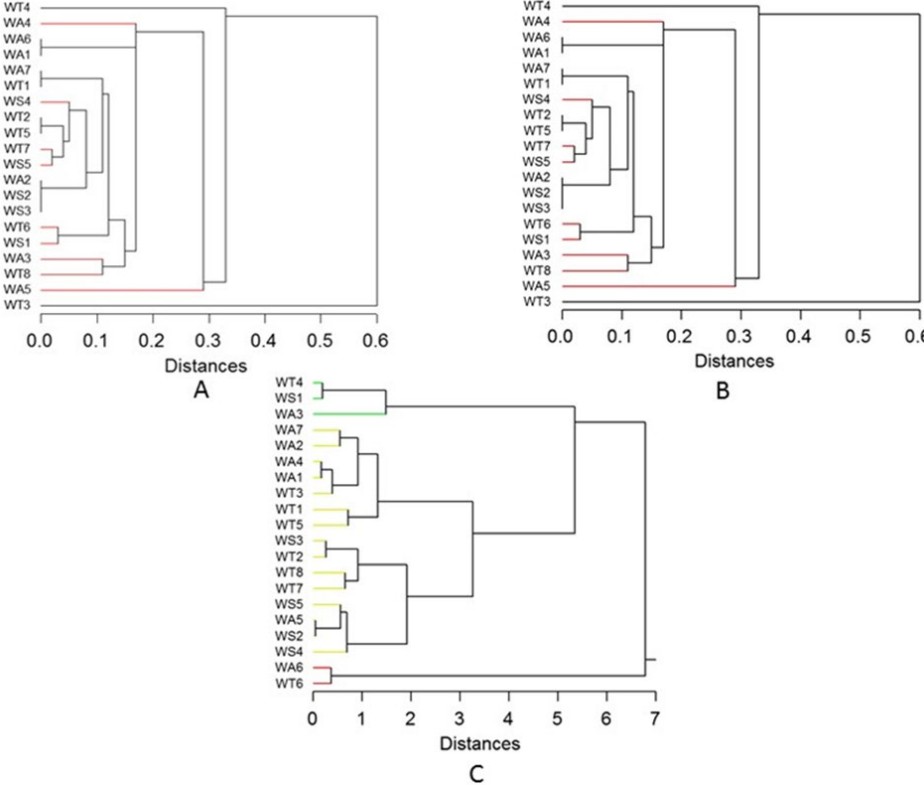

**Figure 2.** Dendrograms represent the results of hierarchical cluster analyses based on stone shape (**A**), leaf shape (**B**), and fruit specification (**C**) of the selected wild olive phenotypes.

### 3.2. Quantitative and Qualitative Analysis of Oil

3.2.1. Olive Oil (OO)

Olive oil (OO) content is a criterion to be envisaged during the varieties selection. As oil content is influenced by olive flesh humidity during the fruit-ripening period, this index was expressed as a percent of fresh matter. The result (Table 2) showed that the average percentage of the OO content varied significantly among phenotypes; most contain more than 18% OO, and this is consistent with the results of studies [46–48]. Some (WS2, WT4, WA6, WA4) showed a high percentage of 26.47, 26.76, 29.01, 29.3%, respectively.

**Table 2.** Percentage of OO in the studied wild olive phenotypes based on the fresh matter weight (*n* = 3).

| Phenotype Code | The Average Percentage of OO in 1st Season | The Average Percentage of OO in 2nd Season | The Average of OO in Two Seasons |
|---|---|---|---|
| WS1 | 21.12 ± 0.16 | 21 ± 16 | 21.06 ± 0.50 |
| WS2 | 26.50 ± 0.02 | 26.33 ± 0.57 | 26.42 ± 0.29 |
| WS3 | 20.16 ± 0.11 | 21.05 ± 0.09 | 20.6 ± 0.13 |
| WS4 | 20.26 ± 0.20 | 21 ± 0.09 | 20.52 ± 0.27 |
| WS5 | 18.22 ± 0.06 | 19 ± 0.12 | 18.61 ± 0.03 |
| WA1 | 12.04 ± 0.05 | 12 ± 0.13 | 11.97 ± 0.02 |
| WA2 | 10.53 ± 0.13 | 10.3 ± 0.51 | 10.43 ± 0.29 |
| WA3 | 11.74 ± 0.16 | 11.33 ± 0.51 | 11.73 ± 0.22 |
| WA4 | 29.27 ± 0.25 | 29.38 ± 0.53 | 29.31 ± 0.40 |
| WA5 | 17.81 ± 0.05 | 17.62 ± 0.51 | 17.73 ± 0.31 |
| WA6 | 29.01 ± 0.08 | 29.02 ± 0.12 | 29.01 ± 0.04 |
| WA7 | 19.04 ± 0.22 | 18.42 ± 0.51 | 18.74 ± 0.19 |
| WT1 | 16.03 ± 0.28 | 16.03 ± 0.12 | 16.01 ± 0.14 |
| WT2 | 12.11 ± 0.22 | 12.22 ± 0.51 | 12.21 ± 0.38 |
| WT3 | 9.17 ± 0.04 | 10.73 ± 0.11 | 9.61 ± 0.04 |
| WT4 | 27.02 ± 0.07 | 26.66 ± 0.52 | 26.76 ± 0.29 |
| WT5 | 10.50 ± 0.05 | 11.46 ± 0.50 | 11.01 ± 0.25 |
| WT6 | 12.41 ± 0.05 | 14.04 ± 0.06 | 13.71 ± 0.03 |
| WT7 | 9.48 ± 0.44 | 10.41 ± 0.52 | 10.02 ± 0.48 |
| WT8 | 22.03 ± 0.27 | 21.62 ± 0.55 | 21.71 ± 0.23 |
| C.V | 1.12 | 2.71 | 1.91 |
| LSD 0.05 | - | - | 0.62 |

The percent of OO extracted during two seasons showed a positive correlation with high significance (r = 0.99) (Table 3).

**Table 3.** Correlations of OO during two seasons (*n* = 3).

| Correlations | The Average Percentage of OO in the 1st Season | The Average Percentage of OO in the 2nd Season |
|---|---|---|
| The average percentage of OO in 1st season. | 1.00 | |
| The average percentage of OO in 2nd season. Pearson Correlation | 0.995 * | 1.00 |

(*) Correlation is significant at ($p < 0.01$).

3.2.2. Fatty Acid Composition

The fatty acid composition is an important quality parameter and authenticity indicator of OO, and a genetic trait that is closely related to the cultivars [49–53] rather than the place of collection, maturity index, and other edaphic factors [54]. Previously, it has been used by several authors as a parameter for oil evaluation [55,56].

The fatty acid evaluation was performed on the OO following the usual product analyses. As can be seen in Table 4, the fatty acid composition of the studied oils is variable

from one phenotype to another. Whereas, the results showed that the percentage of Palmitic fatty acid ranged between 16.15 and 11.2%, Stearic 4.39 and 2.20%, Arachidic 0.88 and 0.36%, Palmitoleic 1.47 and 0.51%, Oleic 77.4 and 62.22%, Linoleic 18.83 and 5.6%, Linolenic 2.08 and 0.19%.

**Table 4.** The composition of fatty acids (%) of the OO wild olive phenotypes (*n* = 3).

| Phenotype Code | Palmitic (C16:0) * | Stearic (C18:0) * | Arachidic (C20:0) * | Palmitoleic (C16:1) * | Oleic (C18:1) * | Linoleic (C18:2) * | Linolenic (C18:3) * | O/L Ratio ** |
|---|---|---|---|---|---|---|---|---|
| WS1 | 13.81 ± 0.57 [b] | 3.23 ± 1.01 [a] | 0.47 ± 0.11 [c] | 0.76 ± 0.11 [b] | 64.91 ± 1.00 [d] | 15.24 ± 1.00 [b] | 1.29 ± 0.60 [b] | 4.26 ± 1.00 |
| WS2 | 14.32 ± 1.01 [a] | 3.14 ± 1.05 [a] | 0.49 ± 0.11 [c] | 0.77 ± 0.11 [b] | 66.05 ± 0.50 [c] | 13.52 ± 1.00 [b] | 1.23 ± 0.60 [b] | 4.89 ± 0.70 |
| WS3 | 12.96 ± 0.71 [b] | 3.17 ± 1.03 [a] | 0.45 ± 0.10 [c] | 0.78 ± 0.12 [b] | 63.76 ± 1.10 [d] | 17.12 ± 1.00 [a] | 1.31 ± 0.68 [b] | 3.72 ± 1.00 |
| WS4 | 13.32 ± 1.02 [b] | 2.77 ± 1.10 [b] | 0.43 ± 0.13 [c] | 0.79 ± 0.12 [b] | 62.22 ± 1.40 [d] | 18.83 ± 1.00 [a] | 1.26 ± 0.60 [b] | 3.30 ± 1.20 |
| WS5 | 13.53 ± 1.03 [b] | 2.85 ± 1.05 [b] | 0.44 ± 0.17 [c] | 1.19 ± 0.4⁶ [a] | 66.15 ± 1.10 [c] | 14.12 ± 1.60 [b] | 1.24 ± 0.90 [b] | 4.68 ± 1.30 |
| WA1 | 11.22 ± 1.11 [c] | 3.05 ± 1.10 [a] | 0.82 ± 0.14 [a] | 0.51 ± 0.14 | 77.41 ± 1.10 [a] | 5.98 ± 1.00 [e] | 0.28 ± 0.10 [c] | 12.94 ± 1.00 |
| WA2 | 16.15 ± 0.92 [a] | 2.98 ± 1.01 [b] | 0.88 ± 0.11 [a] | 1.35 ± 0.71 [a] | 66.79 ± 0.90 [c] | 11.18 ± 1.00 [c] | 0.19 ± 0.10 [c] | 5.97 ± 0.90 |
| WA3 | 15.99 ± 1.21 [a] | 2.20 ± 1.05 [b] | 0.68 ± 0.10 [b] | 1.47 ± 0.75 [a] | 67.11 ± 1.10 [c] | 11.76 ± 1.00 [c] | 0.23 ± 0.15 [c] | 5.70 ± 1.00 |
| WA4 | 15.67 ± 1.10 [a] | 2.85 ± 1.04 [b] | 0.44 ± 0.10 [c] | 0.75 ± 0.11 [b] | 68.45 ± 1.00 [b] | 10.51 ± 1.00 [c] | 0.23 ± 0.10 [c] | 6.51 ± 1.00 |
| WA5 | 14.85 ± 1.11 [a] | 2.83 ± 0.90 [b] | 0.46 ± 0.14 [c] | 0.76 ± 0.11 | 74.87 ± 1.00 [a] | 5.60 ± 1.00 [e] | 0.28 ± 0.10 [c] | 13.36 ± 1.00 |
| WA6 | 14.93 ± 1.03 [a] | 2.49 ± 0.85 [b] | 0.36 ± 0.12 [c] | 1.08 ± 0.57 [a] | 66.74 ± 1.10 [c] | 13.40 ± 1.00 [b] | 0.56 ± 0.10 [c] | 4.98 ± 1.00 |
| WA7 | 13.82 ± 1.14 [b] | 3.13 ± 1.10 [a] | 0.59 ± 0.13 [b] | 1.34 ± 0.71 [a] | 69.11 ± 1.00 [b] | 9.90 ± 1.00 [d] | 2.08 ± 0.60 [a] | 6.98 ± 1.00 |
| WT1 | 14.11 ± 0.99 [a] | 3.01 ± 1.20 [a] | 0.53 ± 0.11 [b] | 0.88 ± 0.10 [b] | 65.51 ± 0.90 [d] | 13.40 ± 1.00 [b] | 1.38 ± 0.70 [b] | 4.89 ± 0.90 |
| WT2 | 13.89 ± 0.99 [b] | 3.01 ± 1.01 [a] | 0.62 ± 0.11 [b] | 0.81 ± 0.10 [b] | 64.99 ± 1.00 [d] | 14.77 ± 1.00 [b] | 1.65 ± 0.80 [b] | 4.40 ± 1.00 |
| WT3 | 13.18 ± 1.10 [b] | 3.40 ± 0.82 [a] | 0.53 ± 0.11 [b] | 0.84 ± 0.10 [b] | 66.39 ± 1.00 [c] | 13.52 ± 1.00 [b] | 1.63 ± 0.80 [b] | 4.91 ± 1.00 |
| WT4 | 13.04 ± 1.12 [b] | 2.60 ± 1.01 [b] | 0.36 ± 0.10 | 0.99 ± 0.06 [b] | 67.81 ± 1.00 [c] | 13.93 ± 1.00 [b] | 1.08 ± 0.60 [b] | 4.87 ± 1.00 |
| WT5 | 14.15 ± 1.03 [a] | 4.39 ± 1 [a] | 0.63 ± 0.13 [b] | 1.17 ± 0.09 [a] | 67.78 ± 1.00 [c] | 9.85 ± 1.00 [d] | 1.31 ± 0.70 [b] | 6.88 ± 1.00 |
| WT6 | 13.65 ± 1.04 [b] | 2.79 ± 1.03 [b] | 0.46 ± 0.16 [c] | 1.18 ± 0.09 [a] | 67.52 ± 1.00 [c] | 12.4 ± 1.00 [c] | 1.50 ± 0.80 [a] | 5.44 ± 1.00 |
| WT7 | 15.17 ± 1.04 [a] | 2.51 ± 1.11 [b] | 0.46 ± 0.11 [c] | 1.25 ± 0.13 [a] | 63.43 ± 1.00 [d] | 15.53 ± 1.00 [b] | 1.25 ± 0.70 [b] | 4.08 ± 1.00 |
| WT8 | 13.01 ± 1.10 [b] | 2.43 ± 1.02 [b] | 0.39 ± 0.11 [c] | 0.67 ± 0.13 [b] | 66.68 ± 0.50 [c] | 15.13 ± 1.00 [b] | 1.16 ± 0.60 [b] | 4.41 ± 0.70 |
| C.V | 6.62 | 30.13 | 17.61 | 36.10 | 1.42 | 8.21 | 32.52 | - |
| LSD 0.05 | 1.55 | 1.47 | 0.15 | 0.60 | 1.58 | 1.76 | 0.61 | - |

(*) The means with different letters in parentheses ([a,b,c,d,e]) for the same quality parameter indicate LSD at the 0.05 confidence level. (**) O/L, oleic acid/linoleic acid ratio.

The results also showed significant differences at the 0.05 LSD for the percentage of fatty acids composition.

The degree of correlation between the fatty acids included in the composition of OO according to the level of the correlation and its significance were different (Table 5).

**Table 5.** The correlation between fatty acids (%) of the OO wild olive phenotypes (*n* = 3).

|  | Palmitic | Stearic | Arachidic | Palmitoleic | Oleic | Linoleic | Linolenic |
|---|---|---|---|---|---|---|---|
| Palmitic | 1 | | | | | | |
| Stearic | 0.13 | 1.00 | | | | | |
| Arachidic | 0.08 | 0.16 | 1.00 | | | | |
| Palmitoleic | 0.52 * | −0.17 | 0.32 | 1.00 | | | |
| Oleic | −0.23 | 0.06 | 0.34 | −0.21 | 1.00 | | |
| Linoleic | −0.06 | −0.18 | −0.48 * | −0.06 | 0.91 ** | 1.00 | |
| Linolenic | −0.34 | 0.43 | −0.39 | −0.23 | −0.55 * | 0.59 ** | 1.00 |

(*) Indicates that they are significantly different (*p* < 0.05). (**) Indicates that they are significantly different (*p* < 0.01).

The table shows there is a high correlation between the two fatty acids (Oleic/Linoleic) r = 0.91 and (Linolenic/Linoleic) r = 0.59. However, there is a negative correlation between Oleic/Linolenic (r = −0.55) and Linolenic/Arachidic (r = −0.48).

Monounsaturated fatty acids have great importance due to their nutritional implication and effect on the oxidative stability of oils [57].

Oleic acid, which is the main monounsaturated fatty acid [58], has an important role as it forms 55 to 83% of the fatty acid composition [59]. It is used as an excipient in pharmaceuticals, and the diet is linked with a reduction in the risk of coronary heart disease (CHD) [58,59] and is widely recommended to replace a similar amount of saturated fat without increasing the total number of daily calories [60–62]. The proportion of oleic acid within the fatty acid composition plays an important role in determining the quality of the

oil as olive oil that contains oleic acid by more than 55% is classified as an excellent virgin olive [63]. Based on proven beneficial health effects, it is recommended to substitute other lipids with oleic acid [64], Moreover, it gives olive oil anti-oxidation properties [65] and the ability to bear high temperatures during cooking.

Oleic acid was found in a wide range in the selected phenotypes, but not less than 62.22% as in WS4. Two phenotypes (WA1, WA5) express a high percentage of oleic acid (77.4%, 74.81%, respectively). This is higher than the higher percentage (71.9%) in Picual grown in Northwestern Argentina [66] and gives great importance to these two phenotypes as the higher the percentage of oleic acid, the higher its quality [67].

The content of stearic acid, another important saturated acid, is within the range of 2.43% in WT8 and 4.39% in WT5. For the arachidic acid, all the studied phenotypes showed values lower than the limit of 0.6% established for the OO except for the WA1, WA2, WA3, WT2, and WT5, which showed somewhat higher values of 0.82%, 0.88%, 0.68%, 0.62%, and 0.63%, respectively.

Polyunsaturated fatty acids, such as linoleic acid, are very important for human nutrition. However, these fatty acids are negatively correlated to the stability of OO as it is much more susceptible to oxidation [68,69]. The phenotypes WS4, WS3, and WT7 showed the highest percentage of 18.83%, 17.12%, and 15.53%, respectively, whereas the lowest percentage was found in WA1 and WA5 of 5.98 and 5.6%, respectively.

Given the important role that oleic acid plays when evaluating the quality of OO, the cluster analyses of the percentage of extracted OO and the content of oleic acid are performed and the result is presented in Figure 3. The first group is WA1, WA5, characterized by high oleic acid (74.87–77.4%) and the extracted OO ranges (11.9–17.73%). The second group has two subgroups; the first subgroup is WT3, WA2, WT5, and WA3, characterized by a moderate content of oleic acid (66.39–67.78%) and the extracted OO range of 9.61 to 20.6%. The second subgroup is WT6, WT2, WT7, WT1, WS5, WA7, WT8, WS1, WS3, and WS4, characterized by a moderate content of oleic acid (62.22–69.11%) and the extracted OO range of 10 to 21.06%. The third group (WS2, WT4, WA6, and WA4) is characterized by lower content of oleic acid (66.05–68.45%) and a high of extracted OO that ranges from 20.6 to 29.3%.

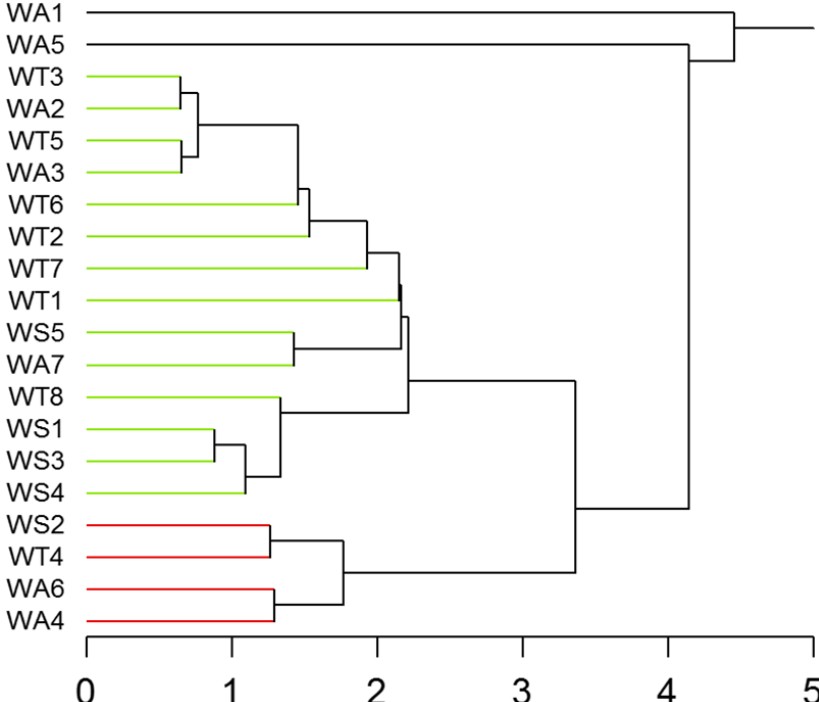

**Figure 3.** The dendrogram represents the results of hierarchical cluster analyses based on the percentage of extracted OO and the content of oleic acid.

Jaccard's similarity coefficient was used to gauge the similarity of the phenotypes according to the following quantitative and chemical morphological properties (leaf, seed, fruit, flower clusters, flesh, OO) in Table 6.

**Table 6.** The similarity of the phenotypes according to quantitative and chemical morphological properties.

|  | WS1 | WS2 | WS3 | WS4 | WS5 | WA1 | WA2 | WA3 | WA4 | WA5 | WA5 | WA7 | WT1 | WT2 | WT3 | WT4 | WT5 | WT6 | WT7 | WT8 |
|---|---|---|---|---|---|---|---|---|---|---|---|---|---|---|---|---|---|---|---|---|
| WS1 | 1.00 | | | | | | | | | | | | | | | | | | | |
| WS2 | 0.44 | 1.00 | | | | | | | | | | | | | | | | | | |
| WS3 | 0.78 | 0.33 | 1.00 | | | | | | | | | | | | | | | | | |
| WS4 | 0.44 | 0.78 | 0.56 | 1.00 | | | | | | | | | | | | | | | | |
| WS5 | 0.56 | 0.67 | 0.67 | 0.89 | 1.00 | | | | | | | | | | | | | | | |
| WA1 | 0.56 | 0.44 | 0.56 | 0.33 | 0.44 | 1.00 | | | | | | | | | | | | | | |
| WA2 | 0.56 | 0.44 | 0.56 | 0.33 | 0.44 | 0.78 | 1.00 | | | | | | | | | | | | | |
| WA3 | 0.78 | 0.56 | 0.56 | 0.44 | 0.56 | 0.78 | 0.78 | 1.00 | | | | | | | | | | | | |
| WA4 | 0.44 | 0.44 | 0.44 | 0.22 | 0.33 | 0.78 | 0.67 | 0.56 | 1.00 | | | | | | | | | | | |
| WA5 | 0.67 | 0.56 | 0.56 | 0.56 | 0.67 | 0.56 | 0.56 | 0.67 | 0.44 | 1.00 | | | | | | | | | | |
| WA6 | 0.33 | 0.67 | 0.44 | 0.67 | 0.78 | 0.44 | 0.33 | 0.44 | 0.44 | 0.44 | 1.00 | | | | | | | | | |
| WA7 | 0.33 | 0.22 | 0.56 | 0.44 | 0.44 | 0.22 | 0.11 | 0.11 | 0.11 | 0.22 | 0.33 | 1.00 | | | | | | | | |
| WT1 | 0.33 | 0.22 | 0.44 | 0.22 | 0.33 | 0.44 | 0.56 | 0.33 | 0.33 | 0.11 | 0.33 | 0.44 | 1.00 | | | | | | | |
| WT2 | 0.22 | 0.11 | 0.44 | 0.22 | 0.33 | 0.56 | 0.56 | 0.44 | 0.56 | 0.44 | 0.33 | 0.33 | 0.44 | 1.00 | | | | | | |
| WT3 | 0.56 | 0.44 | 0.56 | 0.33 | 0.44 | 0.78 | 0.89 | 0.78 | 0.78 | 0.56 | 0.33 | 0.11 | 0.44 | 0.67 | 1.00 | | | | | |
| WT4 | 0.44 | 0.33 | 0.44 | 0.33 | 0.44 | 0.22 | 0.22 | 0.33 | 0.44 | 0.11 | 0.67 | 0.33 | 0.44 | 0.33 | 0.33 | 1.00 | | | | |
| WT5 | 0.33 | 0.22 | 0.33 | 0.11 | 0.22 | 0.67 | 0.67 | 0.56 | 0.67 | 0.56 | 0.22 | 0.22 | 0.44 | 0.89 | 0.78 | 0.22 | 1.00 | | | |
| WT6 | 0.33 | 0.33 | 0.44 | 0.44 | 0.56 | 0.44 | 0.56 | 0.56 | 0.44 | 0.67 | 0.44 | 0.11 | 0.22 | 0.78 | 0.67 | 0.33 | 0.67 | 1.00 | | |
| WT7 | 0.33 | 0.56 | 0.22 | 0.44 | 0.56 | 0.56 | 0.56 | 0.56 | 0.56 | 0.44 | 0.56 | 0.22 | 0.56 | 0.56 | 0.67 | 0.44 | 0.67 | 0.56 | 1.00 | |
| WT8 | 0.56 | 0.67 | 0.44 | 0.67 | 0.78 | 0.44 | 0.56 | 0.56 | 0.56 | 0.67 | 0.56 | 0.22 | 0.33 | 0.33 | 0.67 | 0.44 | 0.44 | 0.56 | 0.78 | 1.00 |

As expected, our present findings clearly show that oleic acid is the major fatty acid in wild olive oils as in the cultivated olive oils. In addition, all of the studied wild olive trees contained more than 62% oleic acid of the fatty acid composition.

Among the screened oleasters, two (WA6, WA4) stand out for their interesting OO content (29.01, 29.3%, respectively) and fairly good oleic acid content (66.74, 68.45%, respectively). As well, the content of extracted OO showed a positive correlation with high significance during two agricultural seasons. This is very important given that the trees are in the wild state (Figure 4).

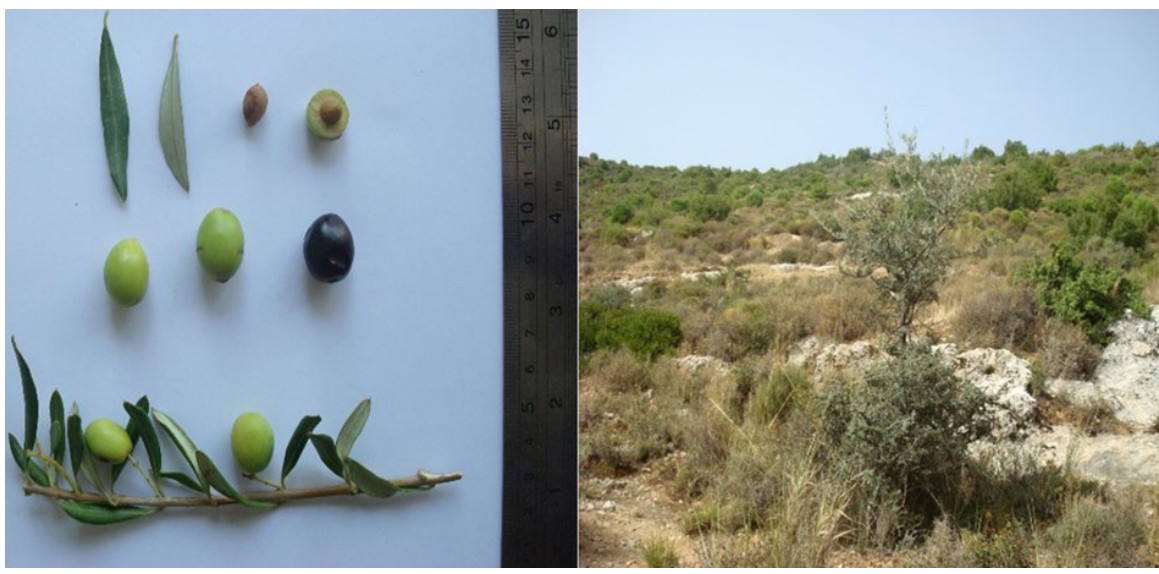

**Figure 4.** Phenotype WA4 (OO 29.3%, oleic 68.45%, flesh weight 75.64%).

Regarding the composition of fatty acids, the significant differences reflect potential genetic diversity apart from the influence of the environment. WA1 had the highest values

of oleic acid (77.4%), whereas both WA4 and WA6 were noteworthy for their high content of OO and oleic acid. Both cluster analysis and Jaccard's similarity coefficient revealed clear relationships between accessions according to their geographical origin. This can be explained by the fact that the spatial juxtaposition has an effect on the similarity. Two wild olive phenotypes (WA4, WA6) were selected as promising since they showed the lowest similarity according to Jaccard's similarity coefficient, in addition to their fruits possessing higher-quality properties (fruit weight 2.16, 3.24 g, flesh 75.83, 86.2, OO% 29.27, 29.01, oleic% 68.45, 66.74, respectively). This enables it to be a very promising introduction to genetic improvement processes.

## 4. Conclusions

In Syria, a wide genetic diversity has been reported for wild olives, which is interesting for the introgression of some traits in breeding programs. However, care must be taken because introducing some beneficial wild traits may have negative effects on other traits. The study revealed the latent potential of the wild varieties of olives that are still present in Syria, not only in terms of the amount of oil extracted but also in terms of the quality of this extracted oil, which qualifies them to be good inputs in the genetic improvement programs for olives. Some of the studied phenotypes showed an average weight of the fruit exceeding 3 g and medium-weight stones with a weight less than 0.4 g. The flesh weight exceeds 70% and reaches 86.66%. Most of them showed a high OO extraction rate at the laboratory scale, while some of them showed to be superior taking into account the harsh environment where they are growing, with some OO values exceeding 29% in addition to the stability of the OO extracted rate during two agricultural seasons. The fatty acid composition was compatible with the commercial specifications of olive oil.

Two phenotypes have been selected as an accession in olive genetic improvement processes, WA4 as an oil accession (OO 29.3%, oleic 68.45%, flesh weight 75.64%), and WA6 as an oil and table accession (fruit weight 3.24 g, OO 29.01%, oleic 66.74%, flesh weight 86.2%).

In future work, these varieties will be further explored for the minor compounds of the oil and in terms of resisting biotic and abiotic stresses.

**Author Contributions:** Conceptualization, R.A.H. and H.H.H.; methodology, R.A.H. and H.H.H.; software, R.A.H.; validation, R.A.H. and H.H.H.; formal analysis, R.A.H.; investigation, R.A.H. and H.H.H.; resources, R.A.H.; data curation, R.A.H.; writing—original draft preparation, R.A.H., H.H.H. and R.B.; writing—review and editing, R.A.H., H.H.H. and R.B. All authors have read and agreed to the published version of the manuscript.

**Funding:** This research received no specific grant from any funding agency, commercial, or not-for-profit sectors.

**Institutional Review Board Statement:** Not applicable for studies not involving humans or animals.

**Informed Consent Statement:** Not applicable.

**Data Availability Statement:** The study did not report any data.

**Acknowledgments:** This work was supported by the biodiversity project in Syria: Biodiversity. The authors would like to extend their sincere appreciation to the staff of the laboratory of fruit physiology at the General Commission of Scientific Agricultural Research (GCSAR) for the help in oil samples analyses.

**Conflicts of Interest:** The authors declare no conflict of interest.

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
