# Peer review of "Characteristics of Some Wild Olive Phenotypes (Oleaster) Selected from the Western Mountains of Syria"

_sustainability, doi:10.3390/su14095151_

Round 1

Reviewer 1 Report

The Authors present studies of evaluation of some technological and production specifications of selected wild olive phenotypes (oleaster) in Syria.

Please change the title, because it is difficult to find some technological and production specification content in the text. If yes, please highlight it in the materials and methods.

For me, the manuscript looks like "looking for new olive origin" and an evaluation of the most important properties (it is not wrong, especially due to biodiversity needs, but the title suggests something else).

The manuscript can be considered for publication after minor revision. 

Author Response

Thank you very much for the effort to review the manuscript.

All comments and notes have been  taken as well as the title had changed to

Characteristics of some wild olive phenotypes (oleaster) selected from the eastern mountains of Syria

Reviewer 2 Report

line 10:  ana - lyses ?

line 40: is

line 54: ...are still existed...

line 109: (more than 20%)

line 145: ...wild olive varieties...

line 152: ?? fresh

line 153: 3,24 g (WA6) ?? .......... WA6, WS2

line 165: 0,66 g ??

line 207: 2,20% (WA3)?

line 208: 0,51% (WA1)?......  5,60% (WA5) ?

line 211: ...acids...?

line 227: ...forms...

line 238:  WT5

line 244-246: is this paragraph concerned to the linoleic acid ??
The highest % according to the Table 4 showed WS4, WS3, WT7
The lowest % WA1, WA5

line 255: WS2 ?

Author Response

Thank you very much for the effort to review the manuscript.

All notes and comments have been taken.

Reviewer 3 Report

Manuscript Number: sustainability-1682451, titled:

Evaluation of some technological and production specifications of selected wild olive phenotypes (oleaster) in Syria

Review 1 – 4 April 2022

Dear Editor of Sustainability

the argument is interesting but has to be improved. The introduction section has to be better argued. The M&M section has to be completed. The bibliography has to be widely increased and improved and included in the text as required by Sustainability (MDPI). Inaccuracies in the manuscript.

I suggest a major revision

To the Authors (in detail):

  1. the argument is interesting but has to be improved. The introduction section has to be better argued. The M&M section has to be completed. The bibliography has to be widely increased and improved and included in the text as required by Sustainability (MDPI). Inaccuracies in the manuscript.

  1. Introduction section, before to restrict the field to fatty acid composition you have to describe the olive oil composition which is composed with triglycerides (98 - 99%) [1], and minor components such as: sterols [2]; waxes [3]; tocopherols [4]; carotenes and chlorophylls [5]; phenolic compounds [6]; volatiles [7]. Please, find, read and discuss all references I have listed, insert the reference number after each class of compound and do not cumulate all references at the end of the sentence.

 [1] Variation in triacylglycerols of olive oils produced in Calabria (Southern Italy) during olive ripening

Riv. Ital. Sostanze Gr. 91 (4), 221-240 (2014).

[2] Importance of Phytosterols in the Classification of Tunisian Olive Cultivars: Discrimination Between Varieties, Hybrids and Oleasters.

Environmental Science and EngineeringPages 1109 – 1113 (2021) 2nd Euro-Mediterranean Conference on Environmental Integration, EMCEI 2019, Sousse10 October 2019 through 13 October 2019Code 258519.

DOI: 10.1007/978-3-030-51210-1_174

[3] Influence of harvest year and cultivar on wax composition of olive oils.

Eur. J. Lipid Sci. Technol. 115 (5) 549-555 (2013)

DOI: 10.1002/ejlt.201200235.

[4] Characterization of commercial Tunisian monovarietal olive oils produced from autochthonous olive cultivars.

Emirates Journal of Food and Agriculture. 2018. 30(7): 581-591

doi: 10.9755/ejfa.2018.v30.i7.1741

[5] Pigments in Extra-Virgin Olive Oils Produced in Tuscany (Italy) in Different Years.

Foods. 2017 Apr; 6(4): 25. doi: 10.3390/foods6040025

[7] Exploitation of virgin olive oil by-products (Olea europaea L.): phenolic and volatile compounds transformations phenomena in fresh two-phase olive pomace (‘alperujo’) under different storage conditions.

Journal of the Science of Food and Agriculture 102, (6) 2515 – 2525 (2022).

DOI: 10.1002/jsfa.11593

[8] Volatile profiles of extra virgin olive oil, olive pomace oil, soybean oil and palm oil in different heating conditions.

LWT - Food Science and Technology 117(1), 108631 (2020).

https://doi.org/10.1016/j.lwt.2019.108631

  1. Please, before to discuss of volatiles as minor components, find, read, discuss and include also one scientific work/reference for fatty alcohols in olive oil, and one for n-alkanes and n-alkenes in olive oil;

  1. 2.1 sub-section, please, detail, the microclimatic conditions; the age of plants and any other information you have;
  2. 2.1 sub-section, please, do not use full maturity, but indicate the fruit color (Jaen index) or days after blossom, or days after anthesis;
  3. Line 99 separate ca. from 27 and 27 from °C;
  4. In the whole manuscript, when you have to indicate a temperature, separate the numeric value from the symbol: 27 °C and not 27°C;
  5. Line 100, delete the dot after hr;
  6. Line 99 (ml), line 113 (mL), please, be consistent in the whole manuscript: ml or mL? I suggest mL;
  7. Line 130: 0.25 mm is the internal diameter;
  8. Line 130: it is impossible 0.25 m film thickness, please, verify carefully;
  9. Section 3, lines 143-148. Not only the pre-harvest factors influence the quality of an olive oil but also the post-harvest factors. Please, support your statement with some other reference with regard to: Pre-and post-harvest factors and their impact on oil composition and quality of olive fruit
  10. 3.1 sub-section and in the whole manuscript, please, separate the numeric value from the unity of measurement: 3.13 g and not 3.13g;
  11. 3.1 sub-section and in the whole manuscript, before to compare your data you have to specify if the ripening degree was the same for all fruits. Full maturity is not enough;
  12. Line 171 (IOC) please, include the reference in your bibliography;
  13. 3.1 sub-section. There is not a discussion of your data by comparing your results with finding of other authors with regard to the biometric evaluation. This is completely missed. Please, find, read and discuss at least [9-10]:

[9] Morphological cladistic analysis of eight popular Olive (Olea europaea L.) cultivars grown in Saudi Arabia using Numerical Taxonomic System for personal computer to detect phyletic relationship and their proximate fruit composition.

Saudi Journal of Biological Sciences 23 (1) 115-121 (2016).

 https://doi.org/10.1016/j.sjbs.2015.05.008

[10] Biometric evaluation of twelve olive cultivars under rainfed conditions in the region of Calabria, South Italy

Emirates Journal of Food and Agriculture 29 (9) 696-709 (2017

  1. 3.1.2 sub-section, why have you included only these fatty acids? C17:1, C20:1; C22:0, C24:0 are missed;
  2. 3.1.2 sub section, include the sum of unsaturated fatty acids, the sum of saturated fatty acids, the ration UFS/SFA,
  3. 3.1.2 sub-section, the sum of the fatty acid composition has to be 100.0, please, verify;
  4. Table 5, re-arrange the correlation matrix with other data;
  5. Page 9, discuss your data of oleic acid with regard to the human health and the daily intake of olive oil. Here include some references;
  6. The References section is not arranged as required by the Instructions for Authors of Sustainability. Please, use also a recently published paper as a template;
  7. Please, include carefully all my comments and write in blue color or evidence differently all the corrections you will do.

Regards.

Author Response

Thank you very much for the effort to review the manuscript.

Most notes and comments have been taken, as well as the title has been changed 

1 -regarding volatiles 

It is not mentioned in any part of the article

2-17.3.1.2 what have you included only these fatty acids.......

Because these fatty acids are the most important in determining the quality of the oil.

3- Reference has been arranged according to the recently published paper  

Round 2

Reviewer 3 Report

Manuscript Number: sustainability-1682451, titled:

Evaluation of some technological and production specifications of selected wild olive phenotypes (oleaster) in Syria

Review 2 – 14 April 2022

Dear Editor of Sustainability

the argument is interesting but the manuscript has to be improved. The Authors have partially included a response to the comments of my rev-1. The introduction section has to be better argued. The M&M section has to be completed. Inaccuracies in the manuscript. The bibliography has to be increased and deeply revised by Authors because there is a lot of inaccuracies and it is not written as required by the Instructions for Authors of Sustainability.

To the Authors (in detail):

  1. The argument is interesting but the manuscript has to be improved. You have partially included a response to the comments of my rev-1. The introduction section has to be better argued. The M&M section has to be completed. Inaccuracies in the manuscript. The bibliography has to be increased and deeply revised because there is a lot of inaccuracies and it is not written as required by the Instructions for Authors of Sustainability. Please, read carefully the instructions for authors, read the template and also use some recently published paper as a template.

  1. Introduction section, you have listed only a part of the compounds existing in a olive oil. In my rev-1 I have asked you to find some reference with regard fatty alcohols [29], n-alkanes and n-alkenes  [30] but you have not found references regarding to these molecules. For this reason now I suggest you two references to include before volatiles. In addition you have proposed your manuscript to Foods, but only one reference of this journal you have included in your bibliography; now I suggest you one reference more of Foods  [30]. Please, find, read and discuss the references I have listed, insert the reference number after each class of compound and do not cumulate all references at the end of the sentence.

 [29] The effects of cultivar and harvest year on the fatty alcohol composition of olive oils from Southwest Calabria (Italy).

Grasas  Aceites 65: e011 (2014).

http://dx.doi.org/10.3989/gya.073913

[30] n-Alkanes and n-alkenes in virgin olive oil from Calabria (South Italy): the effects of cultivar and harvest date.

Foods 2021, 10 (2) 290.

https://doi.org/10.3390/foods10020290

  1. Page 2, line 57, separate the second bracket by as;

  1. Line 90: years in small letters;

  1. 1 sub-section, I have asked you do not use full maturity because it is too generic and I asked you to indicate the ripen index with the Jaen index or days after blossom, or days after anthesis. You have written 0-7 but this is not explicative of the ripening index. 0-7 is a range. Zero means olives with a green peel and a green pulp. 7 means olives with black peel and black pulp. If you want to use the Jaen index you have to calculated it with the specific formula on 100 fruits, but if you do not have it, you can indicate the days after blossom or days after anthesis as period of fruits picking;
  2.  1 sub-section, indicate the harvest year/years of your experiment, the system of fruits picking and from how many plants you have obtained fruits (biological sample);
  3. 2 sub-section, line 140, I have written “impossible” because the film thickness is measured as μm and not as mm (one thousand times less). Please, write correctly your values: (60 m length; 0.25 mm internal diameter; 0.25 μm film thickness). Please, verify your data and replace;
  4. 1 sub-section and in the whole manuscript, before to compare your data you have to specify if the ripening degree was the same for all fruits. Full maturity is not enough;
  5. Table 3, first column by right side: 2nd and not 2end. The same some line below and in the whole manuscript;
  6. 1.2 sub-section, why have you included only these fatty acids? C17:1, C20:1; C22:0, C24:0 are missed;
  7. 1.2 sub section, include the sum of unsaturated fatty acids, the sum of saturated fatty acids, the ration UFS/SFA,
  8. 1.2 sub-section, the sum of the fatty acid composition has to be 100.0, please, verify;
  9. Page 9, discuss your data of oleic acid with regard to the human health and the daily intake of olive oil. Here include some references;
  10. The References section is not arranged as required by the Instructions for Authors of Sustainability. There is a lot of mistake, one and often more than one for each reference. Please, use also a recently published paper as a template.
  11. References section: the journal names have to be abbreviated. You can digit on google: …abbreviated.. and also the journal name, or you can digit on google .. journal name abbreviation and you will have the list of the abbreviation of almost all journal;
  12. References section: the first letter of each word of the journal name has to be abbreviated;
  13. References section: the scientific names have to be written by using the binomial nomenclature, for example, Olea europaea has to be italicized (see you refs 3, 7, 11, 12, 17, 18, 28, 35, 49, 50 and so on). Verify the original paper;
  14. References section: ref, 24, 50, 54 and all references, do not write.. & before the name of the last author, please read carefully the instructions for authors and use some recently published paper as a template;
  15. References section: the journal name has to be abbreviated and italicized;
  16. References section, for example, the abbreviation of this journal is: LWT - Food Sci. Technol.;
  17. References section, for example, ref 28 has to be completely re-arranged;
  18. References section: write in bold the year of publication;
  19. references section: the volume number has to be italicized;
  20. references section: the issue number is not required;
  21. Ref 60, delete [Google scholar];
  22. Refs 3, 25, 27, 65, 66 and in the whole section: why Jg and Nr? Please, delete and re-arrange;
  23. Ref 33, write all authors’ names;
  24. Please, include carefully all my comments and write in red color or evidence differently all the corrections you will do.

Regards.

Author Response

To the Authors (in detail):

  1. The argument is interesting but the manuscript has to be improved. You have partially included a response to the comments on my rev-1. The introduction section has to be better argued. The M&M section has to be completed. Inaccuracies in the manuscript. The bibliography has to be increased and deeply revised because there are a lot of inaccuracies and it is not written as required by the Instructions for Authors of Sustainability. Please, read carefully the instructions for authors, read the template, and also use some recently published papers as a template.

-The manuscript has improved according to your valuable comments, and we have included responses almost to all your comments in rev-1 and rev-2

-The bibliography has increased and deeply revised and has been written as required by the Instructions for Authors of Sustainability.

  1. Introduction section, you have listed only a part of the compounds existing in olive oil. In my rev-1 I have asked you to find some references with regard to fatty alcohols [29], n-alkanes, and n-alkenes [30] but you have not found references regarding these molecules. For this reason, now I suggest you two references to include before volatiles. In addition, you have proposed your manuscript to Foods, but only one reference of this journal you have included in your bibliography; now I suggest you reference more of Foods [30]. Please, find, read and discuss the references I have listed, insert the reference number after each class of compound, and do not cumulate all references at the end of the sentence.

[29] The effects of cultivar and harvest year on the fatty alcohol composition of olive oils from Southwest Calabria (Italy). Grasas Aceites 65: e011 (2014). http://dx.doi.org/10.3989/gya.073913

[30] n-Alkanes and n-alkenes in virgin olive oil from Calabria (South Italy): the effects of cultivar and harvest date. Foods 2021, 10 (2) 290. https://doi.org/10.3390/foods10020290.

The references 29, and 30 have been included.

  1. Page 2, line 57, separate the second bracket as;

The modification was made according to the suggestion

  1. Line 90: years in small letters;

The modification was made according to the suggestion

  1. 2.1 sub-section, I have asked you not to use full maturity because it is too generic and I asked you to indicate the ripen index with the Jaen index or days after blossom, or days after anthesis. You have written 0-7 but this is not explicative of the ripening index. 0-7 is a range. Zero means olives with a green peel and a green pulp. 7 means olives with black peel and black pulp. If you want to use the Jaen index you have to calculate it with the specific formula on 100 fruits, but if you do not have it, you can indicate the days after blossom or days after anthesis as a period of fruit picking;

The modification was made according to the suggestion

  1. 2.1 sub-section, indicate the harvest year/years of your experiment, the system of fruit picking and from how many plants you have obtained fruits (biological sample);

The modification was made according to the suggestion

  1. 2.2 sub-section, line 140, I have written “impossible” because the film thickness is measured as μm and not as mm (one thousand times less). Please, write correctly your values: (60 m length; 0.25 mm internal diameter; 0.25 μm film thickness). Please, verify your data and replace;

The modification was made according to the suggestion

  1. 3.1 sub-section and in the whole manuscript, before comparing your data you have to specify if the ripening degree was the same for all fruits. Full maturity is not enough;

The modification was made according to the suggestion

  1. Table 3, first column by right side: 2nd and not 2end. The same some lines below and in the whole manuscript;

The modification was made according to the suggestion

  1. 3.1.2 sub-section, why have you included only these fatty acids? C17:1, C20:1; C22:0, C24:0 are missed;

First, because it is the most important fatty acid in olive oil

Most oil reviews are based on the percentage of these acids

Finally, there is no possibility to analyze other components

  1. 3.1.2 subsection, includes the sum of unsaturated fatty acids, the sum of saturated fatty acids, the ratio UFS/SFA,

We could not include the sum of unsaturated fatty acids, the sum of saturated fatty acids, the ratio UFS/SFA due to the lack of possibilities to analyze other fatty acids )C17:1, C20:1; C22:0, C24:0(, whether saturated or unsaturated

  1. 3.1.2 sub-section, the sum of the fatty acid composition has to be 100.0, please, verify;

The sum has been verified

  1. Page 9, discusses your data on oleic acid with regard to human health and the daily intake of olive oil. Here include some references;

The discussion has been made and some references have included

  1. The References section is not arranged as required by the Instructions for Authors of Sustainability. There are a lot of mistakes, one and often more than one for each reference. Please, use also a recently published paper as a template.

The modification was made according to the suggestion

  1. References section: the journal names have to be abbreviated. You can digit on google: …abbreviated.. and also the journal name, or you can digit on google .. journal name abbreviation and you will have the list of the abbreviation of almost all journals;

The modification has made according to the suggestion

  1. References section: the first letter of each word of the journal name has to be abbreviated;

The modification has made according to the suggestion

  1. References section: the scientific names have to be written by using the binomial nomenclature, for example, Olea europaea has to be italicized (see refs 3, 7, 11, 12, 17, 18, 28, 35, 49, 50 and so on). Verify the original paper;

The modification has been made according to the suggestion

  1. References section: ref, 24, 50, 54 and all references, do not write.. & before the name of the last author, please read carefully the instructions for authors and use some recently published paper as a template;

The modification has made according to the suggestion

  1. References section: the journal name has to be abbreviated and italicized;

The modification was made according to the suggestion

  1. References section, for example, the abbreviation of this journal is LWT - Food Sci. Technol.;

The modification has been made according to the suggestion

  1. References section, for example, ref 28 has to be completely re-arranged;

The modification was made according to the suggestion

  1. References section: write in bold the year of publication;

The modification has been made according to the suggestion

  1. references section: the volume number has to be italicized;

The modification has made according to the suggestion

  1. references section: the issue number is not required;

The modification has made according to the suggestion

  1. Ref 60, delete [Google Scholar];

The modification has been made according to the suggestion

  1. Refs 3, 25, 27, 65, 66, and in the whole section: why Jg and Nr? Please, delete and re-arrange;

The modification has been made according to the suggestion

  1. Ref 33, write all authors’ names;

The modification has been made according to the suggestion

  1. Please, include carefully all my comments and write in red color or evidence differently all the corrections you will do.

Round 3

Reviewer 3 Report

Manuscript Number: sustainability-1682451, titled:

Former: Evaluation of some technological and production specifications of selected wild olive phenotypes (oleaster) in Syria.

Now: Characteristics of some wild olive phenotypes (oleaster) selected from the western mountains of Syria

Review 3 – 20 April 2022

Dear Editor of Sustainability

the argument is interesting and the Authors have included all my comments.

I suggest the publication of this manuscript in the present form.

Regards.
